# Electrical and Structural Properties of Semi-Polar-ZnO/*a*-Al_2_O_3_ and Polar-ZnO/*c*-Al_2_O_3_ Films: A Comparative Study

**DOI:** 10.3390/ma16010151

**Published:** 2022-12-23

**Authors:** Sushma Mishra, Wojciech Paszkowicz, Adrian Sulich, Rafal Jakiela, Monika Ożga, Elżbieta Guziewicz

**Affiliations:** Institute of Physics, Polish Academy of Sciences, Al. Lotników 32/46, 02-668 Warsaw, Poland

**Keywords:** ZnO, strain, microstrain, dislocation density, morphology, SIMS, electrical conductivity

## Abstract

In this work, the properties of ZnO films of 100 nm thickness, grown using atomic layer deposition (ALD) on *a*–(100) and *c*–(001) oriented Al_2_O_3_ substrate are reported. The films were grown in the same growth conditions and parameters at six different growth temperatures (T_g_) ranging from 100 °C to 300 °C. All as-grown and annealed films were found to be polycrystalline, highly (001) oriented for the *c*–Al_2_O_3_ and highly (101) oriented for the *a*–Al_2_O_3_ substrate. The manifestation of semi-polar-(101) and polar (001)–oriented ZnO films on the same substrate provided the opportunity for a comparative study in terms of the influence of polarization on the electrical and structural properties of ZnO films. It was found that the concentration of hydrogen, carbon, and nitrogen impurities in polar (001)–oriented films was considerably higher than in semi-polar (101)–oriented ZnO films. The study showed that when transparent conductive oxide applications were considered, the ZnO layers could be deposited at a temperature of about 160 °C, because, at this growth temperature, the high electrical conductivity was accompanied by surface smoothness in the nanometer scale. On the contrary, semi-polar (101)–oriented films might offer a perspective for obtaining p-type ZnO films, because the concentration of carbon and hydrogen impurities is considerably lower than in polar films.

## 1. Introduction

ZnO is a well-known multifunctional material [1]. A wide and direct band gap (3.37 eV) and high exciton binding energy (60 meV) mean that ZnO has significant potential for applications in light-emitting devices (LED), as well as in photovoltaic and solar cells as a transparent conductive electrode [2,3,4]. According to current knowledge, the presence of native point and structural defects, impurities (such as H, C, N), and associated complexes determine the level of electrical conductivity, as well as the concentration of carriers and their mobility [5,6,7]. In order to correctly design ZnO-based devices, it is necessary to differentiate the impact of structural defects and impurities, as well as native point defects and their complexes, on the electrical properties of the material. It has been shown that electrical properties of ZnO considerably depend on Zn/O-rich conditions, because they influence the formation energy of specific native point defects [1,8,9] However, it is challenging to separate the role of native point defects and structural defects in the electrical properties of ZnO. Information of this kind could be obtained by comparing thin, differently oriented ZnO films deposited under identical growth conditions. The same growth conditions ensure the same formation energy of native point defects, whereas the different orientation of the ZnO layer, obtained by growing the films on differently oriented substrates, gives rise to the formation of different structural defects. Despite a large number of papers that have described the structural and electrical properties of ZnO (e.g., [10,11]), such studies have not been conducted so far. The only recent study was carried out last year, where pairs of ZnO films were simultaneously grown on Si (100) and *a*-Al_2_O_3_ substrates [12]. However, the results were not entirely clear, because the electron mobility in the annealed ZnO/Si films showed extreme values, indicating the formation of 2D electron gas at the ZnO/Si interface. The present study, in which the ZnO deposition was performed on *c*- and *a*-Al_2_O_3_ substrates, was carried out to avoid this issue.

Due to the piezoelectric nature of ZnO, the results of this kind are intriguing, because the polarization-induced electric field separates the electrons and holes, increasing the recombination time and worsening the performance of an optical device [13]. In this way, the electron and hole recombination time influences the quantum efficiency of semiconducting optoelectronic devices. Consequently, crystallographic orientation might have a significant impact on structural, optical, or electrical characteristics. It is worth noting that studies of this kind are rare, because ZnO films have a strong tendency to grow in the polar (001) direction (i.e., with *c*-axis perpendicular to the substrate’s surface), while highly textured semi-polar (101) ZnO films are difficult to prepare and require very specific deposition conditions and substrates [14,15,16].

As such, in the present study, highly textured polar (002) and semi-polar (101) ZnO films were obtained on *a*-Al_2_O_3_ and *c*-Al_2_O_3_ substrates. The films were grown using the ALD technique, which is particularly advantageous for low temperature growth and nanometer-thick films, which are usually highly conformal due to self-limiting surface reactions. Additionally, the ALD layers were pinhole-free, uniform, smooth, and homogeneous. An additional advantage was the possibility to perform ALD growth on large substrates [17,18,19,20].

For the experiments, a series of ZnO films were grown at temperatures of 100, 130, 160, 200, 250, and 300 °C. As shown, this growth temperature influenced the stoichiometry of the ZnO films, changing them from O-rich to Zn-rich when the temperature rose from 100 to 200 °C and at still higher temperatures [21]. We expected that the formation energy of native point defects was accordingly varied [1]. It was expected that acceptor-like defect complexes, such as n∙V_Zn_ or V_Zn_∙*n*H, are readily formed under O-rich conditions, while Zn-rich conditions favored the formation of donor-like complexes, such as *n*Zn_i_∙V_O_ or Zn_i_∙V_O_H. It was found that electrical conductivity depends on the crystallographic quality of the ZnO films, such as orientation, dislocation density, strain, and microstrain. It is demonstrated that the ZnO/c-Al_2_O_3_ films had larger crystallite sizes, lower strains/microstrains, and higher concentrations of impurities than the ZnO/*a*-Al_2_O_3_ films. In turn, the as-grown ZnO/*a*-Al_2_O_3_ films showed higher mobility values than their ZnO/c-Al_2_O_3_ counterparts, which could be related to lower concentrations of impurities; however, for annealed samples, these dependences were not so straightforward.

## 2. Growth Details and Experimental Techniques

A series of thin ZnO (~100 nm) films were deposited on *a*- and *c*-Al_2_O_3_ substrates at six growth temperatures (T_g_): 100 °C, 130 °C, 160 °C, 200 °C, 250 °C, and 300 °C, using the ALD process in a Savannah-100 reactor from Cambridge Nanotech, Cambridge, US.

A double exchange chemical reaction between de-ionized water and diethyl zinc [(C_2_H_5_)_2_Zn], typical for ZnO-AlD growth, was used for the deposition. The ALD procedure was carried out over 1000 cycles with a pulsing time of 20 ms for both precursors, and purging in N_2_ gas for 20 s and 8 s after the deionized water and DEZn pulses, respectively. It is worth noting that the ZnO films on both substrates were concurrently deposited during the same ALD process. All as-grown and annealed films were characterized by X-ray Diffraction (XRD), Secondary Ion Mass Spectrometry (SIMS), Room Temperature (RT, PhysTech GmbH, Moosburg, Germany) Hall effect, UV-Vis/NIR spectrophotometry, and Atomic Force Microscopy (AFM). The film thickness, which was found between 97 and 162 nm, was evaluated by cross-sectional SEM images. Cu Kα1 radiation (λ = 1.5406 Å) was applied in order to perform structural XRD measurements on a Philips (X’pert PRO MPD diffractometer from Malvern PANalytical, Westborough, MA, USA. SIMS with a Cameca IMS 6f microanalyzer (CAMECA, Gennevilliers Cedex, France) was used to determine the concentrations of H, C, and N in the ZnO films. All three elements were examined as negative secondary ions; specifically, ^1^H, ^12^C, and ^14^N^16^O for hydrogen, carbon, and nitrogen [22]. Because nitrogen atoms have a low ionization yield due to their high (>14 eV) ionization potential, the NO cluster was detected instead of N, with a high mass resolution m/Δm = 1000. The Hall effect at RT was measured on square (1 × 1 cm^2^) samples in the Van der Pauw configuration using an RH2035 PhysTech88 system (PhysTech GmbH, Moosburg, Germany) equipped with a 0.43 T permanent magnet. A Kurt Lesker’s PVD75 e-beam evaporation system was used to deposit the Ti/Au films for ohmic contacts. All of the films were subjected to 3-min Rapid Thermal Annealing (RTA) in an O_2_ atmosphere using an Accu Thermo AW-610 system from Allwin21 Corporation., (Morgan Hill, CA, USA). High-power metal halide lamps were used for annealing and a temperature of 800 °C was achieved in 20 s, whereas cooling down was reached in 300 s. An Agilent Technologies’ Carry 5000 UV-Vis/NIR (Blacksburg, Santa Clara, CA, USA) spectrophotometer with a PbS detector was used to measure absorption spectra. The surface roughness was measured with a ScanAsyst-AIR (Bruker, Santa Barbara, CA, USA) probe using an AFM Dimension Icon by Bruker in the Peak Force Tapping mode (tip radius of 2 nm). Images of 200 × 200 nm and 10 × 10 µm were captured with 512 × 512 measurement points under ambient conditions.

## 3. Experimental Results

### 3.1. AFM

The root mean square (RMS) of surface roughness measured via AFM shows that surface morphology of ZnO deposited at the same T_g_ is similar for the two substrates (Figure 1). The surface roughness had a maximum lowest (100–130 °C) and highest (300 °C) growth temperature, while having a minimum at (~160–200 °C). After post-growth annealing, roughness decreased in ZnO/*c*-Al_2_O_3_ (except at T_g_ = 250 °C) and slightly increased in ZnO/*a*- Al_2_O_3_, except the lowest (100, 130 °C) and highest (300 °C) T_g_.

### 3.2. XRD

X-ray diffraction patterns showed that the as-grown and annealed ZnO films exhibited strong or highly preferred orientation, as indicated by the intensities of either 100 or 101 reflections (Figure 2). The experimental X-ray diffraction peaks matched well to the ZnO data from a representative JCPDS, record 36–1451 [23]. Namely, the matching was observed for the reflections corresponding to the (001) and (if relevant) (101) plane orientations.

Even though films of both types, ZnO/*a*-Al_2_O_3_ and ZnO/*c*-Al_2_O_3_, were grown using the same deposition parameters at six T_g_ (100–300 °C) by the ALD technique, the as grown ZnO/*c*-Al_2_O_3_ films were mostly (001)-oriented (Figure 2a), whereas as-grown ZnO/*a*-Al_2_O_3_ films mostly show the semi-polar (101) orientation (Figure 2c). This tendency was even more pronounced for samples subjected to 3-min RTA in O_2_, when all minor peaks disappeared, only those representing a single orientation remained (Figure 2b,d).

For ZnO/*c*-Al_2_O_3_, all as-grown films were preferentially (001)-oriented (Figure 2a), but it should be noted that, at T_g_ = 200 and 250 °C, the 002 peak showed a shoulder on the left side [24]. That could be related to lattice distortion, which was relaxed after a short RTA (Figure 2b). After annealing, only the 002 peak was observed at all growth temperature ranges (Figure 2b), but with enhanced intensity. This implies that the highly textured (001)-oriented ZnO films were obtained on the *c*-Al_2_O_3_ substrate for all growth temperature ranges.

For ZnO/*a*-Al_2_O_3_, all as grown films consisted of two orientations, (101) and (001), at all growth temperatures, 100–300 °C, but the (101) orientation was the preferred one (Figure 2c). However, after annealing, only the 101 reflection, corresponding to the semi-polar (101) film orientation, appears, but with a higher intensity. The only exception was the film deposited at 160 °C, where the 002 peak corresponding to the (001) plane was more pronounced. In this way, after RTA, highly textured (101)-oriented films were achieved for all growth temperature ranges, except 160 °C (Figure 2d).

Interesting information could be extracted from XRD relative peaks intensity. Namely, in as grown ZnO/*c*-Al_2_O_3_, the 002 peak became successively intensified with increasing T_g_ up to 160 °C, and, beyond this, at T_g_ between 200 and 250 °C, the relative intensity of the 002 peak decreased, and was again intensified at 300 °C (see Appendix A). The same tendency was observed after annealing, where the highest intensity of the 002 peak appeared at a growth temperature of 160 °C (Appendix A).

In as grown ZnO/*a*-Al_2_O_3_, the relative intensity of peak corresponding to preferred (101) orientation increased, though only up to T_g_ = 250 °C, with the exception of T_g_ = 160 °C, where the polar (001) orientation was more intensive than the semi-polar (101) one (see Appendix A). Although the relative preferential intensity of the 101 peak increased after annealing as compared to the as grown films, the dominance of the 002 peak in ZnO/*a*-Al_2_O_3_ could still be observed at T_g_ = 160 °C. Surprisingly, the intensity variation of the polar 002 peak corresponding to the (001) plane (polar) with successive T_g_ was similar to the as grown films, both ZnO/*a*-Al_2_O_3_ and ZnO/*c*-Al_2_O_3_, whereas, after annealing, the (001) orientation disappeared in the ZnO/*a*-Al_2_O_3_ films. The only exception was T_g_ = 160 °C, where the (001) orientation still dominated (Figure 2d). Despite the fact that all the films were oriented along the preferred direction, highly textured films, showing only a single, (001) or (101), orientation, were only achieved after annealing (Figure 2b,d).

Peak switching, meaning a change of orientation between polar to semi-polar, or vice versa, with increasing T_g_, detected for the as-grown ZnO/*a*-Al_2_O_3_ films, has been previously reported for ZnO/Si and ZnO/*a*-Al_2_O_3_ films [12,14] and was attributed to the premature dissociation of DEZn during ALD growth, which can occur at temperatures between 160–250 °C, and to the O/Zn-rich conditions that together influence the preferential directions of the ZnO film growth [12,14,21]. When ZnO films are grown at a low temperature of 100 °C, i.e., under O-rich conditions, they typically develop as a mixture of polar, non-polar, and semi-polar orientations, depending on the orientation of the substrate. While, under Zn-rich conditions, i.e., at T_g_ of 200 °C and higher, ZnO layers grow with the *c*-axis perpendicular to the substrate [14,15,16]. However, for ZnO/*a*-Al_2_O_3_ films, we observed (101) orientation, even for T_g_ = 200, 250, and 300 °C. This implies that growth conditions, together with the thermal expansion coefficient of the film and the substrate, might affect the orientation of the thin ZnO layer.

As presented in Figure 2a–d, the ZnO/*c*-Al_2_O_3_ films do not exhibit any orientation switching, while this phenomenon occurs for the ZnO/*a*-Al_2_O_3_ films at T_g_ = 160 °C, where the 002 peak appears. However, even though all ZnO/*c*-Al_2_O_3_ films are (001)-oriented, a broadening of the 002 peak is observed for T_g_ = 200 °C, and, for T_g_ = 250 °C, a shoulder of this peak is clearly visible, which is a fingerprint of some lattice deformations [24]. Additionally, the relative intensity of the 002 peak shows considerable variations with T_g_, as shown in Appendix A. This denotes a distortion of the crystal lattice, also reflected in surface morphology. The crystallite size, which is directly affected by the above-mentioned substantial structural disorder, decreases towards T_g_ = 200–250 °C, as presented in Figure 3a.

In order to calculate the crystallite size and microstrain, we employed the single-peak analysis technique provided by de Keijser, Langford, Mittemeijer, and Vogels [25,26], which is based on the concept that microstrain broadening is of Gaussian type, while crystallite size broadening is of Lorentzian type. According to this, all diffraction reflections are fitted with the Voigt profile after subtracting instrument broadening, and then two integral breadths, β_L_ and β_G_, corresponding to the Lorentzian and Gaussian contributions, are extracted.

#### 3.2.1. Crystallite Size

The crystallite size in ZnO/*c*-Al_2_O_3_ is 20–70 nm and increases to 40–90 nm after RTA. Similarly, for ZnO/*a*-Al_2_O_3_ films, the crystallites size is 0–30 nm and, after annealing, the observed size is 30–40 nm (Figure 3a).

For all measured films, the average crystallite sizes increased after RTA. This could have been accomplished by accounting for the migration of atoms inside the tiny crystallites during rapid thermal annealing, by diffusion of crystallite boundaries, leading to increase of the crystallite size [27]. The transfer of atoms at crystallite boundaries from one crystallite to another is the process of crystallite development, and the ultimate crystallite size is strongly related to the annealing conditions. The mobility of charge carriers is controlled by the polycrystalline crystallite size; therefore, crystallite size is a key structural characteristic of thin films [9]. The obtained crystallite size values are a few times larger than those reported for ZnO/Si films grown by RF sputtering [28], and are slightly larger than the grain sizes reported for ZnO−ALD films with comparable thickness deposited on glass substrate [29].

#### 3.2.2. Dislocation Density

Dislocation density, δ, is a parameter describing the dislocation lines per unit volume, which allows for the evaluation of the structural quality of single crystals [12,30]. It is estimated using the crystallite size using the well-known relation δ = $1/D^{2}$. In polycrystalline materials, the above formula has also been used for approximations of the dislocation density [12,30]. In fact, the δ value is determined by the size of the crystallites and can be used as a parameter by which to describe the number of structural imperfections and the structural quality of the film, making it a useful tool for comparing various layers. Following this approach, we calculated δ for all of the studied ZnO films, taking the obtained values into consideration with the aforementioned limitations.

In the present samples, the δ value for the as-grown ZnO/*c*-Al_2_O_3_ films was evaluated as 1 × 10^10^−6 × 10^11^ lines/cm^2^. It decreased with successive growth temperatures up to T_g_ = 160 °C. When T_g_ > 160 °C, δ reached its maximum near T_g_ = 200 °C, and then fell again to the lowest value of 1 × 10^10^ lines/cm^2^ (Figure 3b). For as-grown ZnO/*a*-Al_2_O_3_ films, the dislocation density lies in a range between 1 and 6 × 10^11^ lines/cm^2^. It should be noted that even though δ for both as-grown films showed variations in growth temperature, with higher values in the case of the ZnO/*a*-Al_2_O_3_ films compared to ZnO/*c*-Al_2_O_3_. The only exception was for T_g_ = 160 °C, when the orientation switching in the ZnO/*a*-Al_2_O_3_ films was found.

The dislocation density value decreased by one order of magnitude, to 10^10^ lines/cm^2^, after annealing. On the other hand, both types of annealed films showed that the value of dislocation density was almost the same, except when T_g_ = 250–300 °C, where slightly higher δ in ZnO/*a*-Al_2_O_3_ films was observed. This implies that the high dislocation density in ZnO/*a*-Al_2_O_3_ films thermodynamically hinders the usual post-annealing improvement in average crystallites sizes.

#### 3.2.3. Strain and Micro-Strain

In thin polycrystalline films, total strain is a combination of microstrain and macrostrain that can be differentiated via the length scale of their origin [31,32]. Microstrain is an outcome of changes at a short length scale; for example, within the individual crystallite or between adjacent crystallites. Extended defects such as a network of dislocations, twin boundaries, etc., can influence the microstrain via local variation in the lattice parameters. In contrast, macrostrain manifests itself as the lattice parameter variation/changes at a macroscopic scale, namely over a large number of crystallites or, sometimes, even over entire polycrystalline film. These large-scale changes are usually caused by external factors like temperature/pressure changes or composition variations.

For the as-grown ZnO/*c*-Al_2_O films, the strain and microstrain values vary between (~0.15–1.1)% and (1–11 × 10^−3^), respectively, with a maximum of (~1.2%/12 × 10^−3^) observed at T_g_ = 200 °C. On the other hand, for annealed films, the strain/microstrain values are 0.1%–1.2% and 0.5–4 × 10^−3^. Overall, in all T_g_ range the 3 min. RTA treatment does not significantly reduce the strain and microstrain values except when T_g_ = 200 °C, as shown in Figure 3c,d.

For the as-grown ZnO/*a*-Al_2_O_3_ films, the strain and microstrain values are 0.3–0.9% and 3–12 × 10^−3^, respectively. However, after annealing, these values vary between 0.6–0.8% and 1–5 × 10^−3^, respectively. It should be noted that a slight reduction in strain after RTA is observed only for samples grown between 160–300 °C, while, in the samples grown when T_g_~100–130 °C, the strain slightly increases. Furthermore, the microstrain significantly decreases after RTA for all T_g_ ranges (100–300 °C). Finally, it can be stated that, in contrary to the ZnO/*c*-Al_2_O films, the strain behaves differently in respect to the microstrain in annealed ZnO/*a*-Al_2_O_3_ films.

### 3.3. Electrical Properties

RT-Hall measurements were performed to compare electrical parameters such as resistivity, carrier concentration, and mobility for both the polar-ZnO/*c*-Al_2_O_3_ and semi-polar- ZnO/*a*-Al_2_O_3_ films. The resistivity and carrier density and mobility dependence versus T_g_ was similar for both types of as grown (polar/semi-polar) ZnO films; however, the ZnO/*a*-Al_2_O_3_ films showed higher mobility, especially when T_g_ = 160 °C or higher (Figure 4a–c).

After 3 min, RTA carrier density significantly dropped and resistivity increased by three orders of magnitude. The mobility behavior after RTA was quite opposite for both annealed films. In the case of the semi-polar films, it significantly dropped, while it was enhanced in the polar films. A closer look at the mobility versus T_g_ dependence for annealed polar films (Figure 4c) revealed that the mobility is very similar to the microstrain vs. T_g_ dependence, which indicates a link between them.

**Thin ZnO/*c*-Al_2_O_3_ films:** The resistivity of the as-grown films lies in the range of (~1.9–1.6 × 10^−3^) ohm-cm and successively decreases until T_g_ = 250 °C, and then increases at T_g_ = 300 °C. After annealing, the value of resistivity is more pronounced (~1.7–11.1 ohm-cm) and decreases with successive T_g_, though only up to 200 °C. The carrier density (n) in the as-grown films lies in the range of 10^18–20^ cm^−3^, and decreases by up to 2–3 orders of magnitude (n = 10^16–17^ cm^−3^) after RTA. The mobility value of the as-grown films is μ = 3.7–22.7 cm^2^/Vs, (max. at 160 °C and min. at T_g_ = 100 °C) and it is enhanced for all T_g_ ranges (except 160 °C and 250 °C) after annealing. After RTA, the value of mobility lies in the range of μ = 5.9–52 cm^2^/Vs (max. at 130 °C and min.at T_g_ = 250 °C) (see Figure 4c and Table 1).

**Thin ZnO/*a*-Al_2_O_3_ films:** The resistivity of the as-grown films also lie in the range of (~1–10^−3^ ohm-cm) and successively decreases till T_g_ = 200 °C. After 3 min of annealing at 800 °C, the resistivity value increases up to (~0.9–9.6 ohm-cm) with no systematic variation with T_g_ (Figure 4a). The carrier concentration (n) of the as-grown films lies in the range n = 10^18–20^ cm^−3^, being the same as that observed in ZnO/*c*-Al_2_O_3_, but decreases by 1–4 orders of magnitude (n = 10^16–18^ cm^−3^) after RTA. The mobility (µ) values of the as-grown films are between 1.2 and 29.2 cm^2^/Vs, (max. at 160 °C and min. at T_g_ = 100 °C) and decrease after RTA for all T_g_ range, except 130 °C. Mobility (µ) values after RTA are in the range of 3.7–25.2 cm^2^/Vs (max. at 300 °C and min. at T_g_ = 100 °C) (Figure 4c, Table 2).

Overall, only a minor difference is found in terms of resistivity and carrier density in the as-grown ZnO/*a*-Al_2_O_3_ and ZnO/*c*-Al_2_O_3_ films. However, the mobility value is significantly higher in the case of the as-grown ZnO/*a*-Al_2_O_3_ compared to the annealed ZnO/*c*-Al_2_O_3_ films. Interestingly, maximum mobility is observed at T_g_ = 160 °C in both of the as-grown ZnO/*a*-Al_2_O_3_ and ZnO/*c*-Al_2_O_3_ films. It should be noted that, in this case, both films exhibit the same (001) orientation (Figure 2). Resistivity is lowered in (101)-oriented films and mobility is higher [18] in the as-grown ZnO/*a*-Al_2_O_3_ films, which might be due to the switching phenomenon, structural imperfections, and a large lattice mismatch in ZnO/*a*-Al_2_O_3_ as compared to ZnO/*c*-Al_2_O_3_.

On the other hand, by comparing both polar (001) and semi-polar (101)-oriented annealed ZnO/*a*-Al_2_O_3_ and ZnO/*c*-Al_2_O_3_ films, a significant difference in resistivity, carrier concentration, and mobility is observed. In general, the resistivity considerably increases to 1–10 Ωcm, and the carrier concentration considerably decreases (10^16–17^ cm^−3^) after annealing [12] in both types of films. The resistivity value of annealed ZnO/*a*-Al_2_O_3_ films grown at T_g_ = 100 °C, 160 °C, and 300 °C is lower, while it is higher for films grown at T_g_ = 130 °C, 200 °C, and 250 °C compared to ZnO/*c*-Al_2_O_3_ films. A comparison of carrier density (*n*) and mobility (μ) in both annealed films reveals that *n* is higher and μ is lower in ZnO/*a*-Al_2_O_3_ films at $T_{g}\;\leq \;200{\;}^{o}C$, in contrary to $T_{g}\;\geq \;200{\;}^{o}C$, where a decrement in *n* and increment in μ is measured. The difference in n and μ beyond 200 °C might be due to a large difference in crystallites size, which is lower in ZnO/*a*-Al_2_O_3_. As a result, one can say that semi-polar ZnO/*a*-Al_2_O_3_ films show a better conductivity compared to polar ZnO/*c*-Al_2_O_3_ films.

A comparison of the Debye length, L_D_, [9] with the grain size shows that for all of the as-grown films, L_D_ is much smaller than a half crystallite size, D/2. This means that the transport of charge carriers is strongly influenced by potential grain boundary barriers. In the case of annealed films, the L_D_ > D/2 condition is fulfilled only for polar films deposited at 100 and 130 °C. In the latter case, grain boundary scattering is not considered to be dominant.

### 3.4. SIMS

The magnitude of hydrogen, carbon, and nitrogen concentrations were investigated via a secondary ion mass spectrometry depth profile (see Appendix A). Impurities incorporation was different in polar- and semi-polar-oriented ZnO films, which can be seen in Figure 5. Interestingly, after annealing hydrogen, carbon, and nitrogen, impurities were stabilized in the approximate range of (~10^18^–10^19^ at/cm^3^), regardless of the orientation of ZnO films.

**Thin ZnO/*c*-Al_2_O_3_ films:** The H-concentration in the as grown films is 10^20^–10^21^ at/cm^3^ for all growth temperature ranges (100–300 °C), and it continuously decreases with T_g_ (Figure 5a); however, when T_g_ is between 160–300 °C, the H-concentration is about three times higher in ZnO/*c*-Al_2_O_3_ films. After annealing, the H-concentration falls by two orders of magnitude and stabilizes around 10^19^ at/cm^3^ at all T_g_. Similarly, the C-concentration in the as-grown films (Figure 5b) lies in the range 10^20^–10^21^ at/cm^3^; thus, it is similar to hydrogen concentration. On the other hand, annealing does not significantly influence carbon impurity, which is stabilized at the level of 10^20^ at/cm^3^ for all T_g_ ranges. Unintentional N-incorporation (~10^17^–10^18^ at/cm^3^) is also seen in the as-grown films (Figure 5c). It is due to N_2_-gas being used for purging during the ALD growth. All as-grown and annealed polar ZnO/*c*-Al_2_O_3_ films contain 2–4 times more nitrogen than ZnO/*a*-Al_2_O_3_ film. The nitrogen concentration is not affected by the RTA process and still lies in the same range of ~10^18^ at/cm^3^ for all T_g_.

**Thin ZnO/*a*-Al_2_O_3_ films:** The amount of H in the as-grown films is 10^19^–10^21^ at/cm^3^ and successively decreases with increasing growth temperature (Figure 5a). Furthermore, after annealing, the H-concentration decreases by two orders of magnitude and stabilizes around 10^19^ at/cm^3^ at all T_g_. Similarly, carbon concentration in the as-grown films is 10^19^ at/cm^3^, so it is one order of magnitude lower than in ZnO/*c*-Al_2_O_3_ films. RTA does not influence the carbon impurity concentration, which stabilizes at the level of 10^19^ at/cm^3^ at all T_g_ values. The unintentional N-concentration is measured as 10^17^ at/cm^3^ at all T_g_ ranges, and is not affected by RTA.

A comparison of all the as grown films shows that all ZnO/*c*-Al_2_O_3_ films contain impurities (H, C, N) at concentrations higher as compared to those of ZnO/*a*-Al_2_O_3_ films. After the RTP process, the concentration of hydrogen is reduced to the level of 10^19^ at/cm^3^, which is similar in both types of films, whereas the carbon and nitrogen concentrations are not strongly affected and still remain higher in the polar films. One exception occurs at T_g_ = 100 and 200 °C, where carbon concentration in the as grown ZnO/*c*-Al_2_O_3_ films is high (~10^21^ at/cm^3^, Figure 5b) and significantly drops after RTA. The large amount of impurities in the as-grown polar ZnO films may be attributed to structural changes at these specific growth temperatures (see Figure 1a,b).

This implies that the incorporation of carbon and nitrogen impurities into ZnO thin films is not only influenced by growth conditions, but also by the orientation of the film and annealing treatment. The reason behind this might be differences between micro- and macrostrains (>0.1%) and dislocation density that both change with orientation.

## 4. Summary and Discussion

By choosing the substrate and growth conditions, highly textured single (001) and (101)–oriented thin ZnO films of 100 nm thickness were deposited at temperatures belonging to a wide range, 100–300 °C, by atomic layer deposition followed by annealing. The surface morphology for films of both types varies, similarly, with T_g_, showing the lowest roughness of 1–2 nm at T_g_ = 160–200 °C and the highest (~5–7 nm) at 100 and 300 °C. Surface morphology of (101)–oriented ZnO films is not significantly affected by RTA, in contrast to polar (001)–oriented ZnO films, which might be the effect of high strain.

The temperature T_g_ = 160 °C is found especially appropriate for the growth of polar ZnO films because, at this temperature, ZnO grows along the *c*-axis and the respective diffraction peak shows a high intensity for both substrates indicating higher crystalline quality. On the contrary, the 101 diffraction peak is observed only for ZnO/*a*-Al_2_O_3_ films, with the highest intensity for T_g_ = 200 °C (as grown) and 250 °C (annealed), and it is accompanied by a high strain, microstrain, and dislocation density.

The polar (001)–oriented films show larger crystallites, so lower dislocation density, and reveal lower strain and microstrain levels than the semi-polar films. Furthermore, carbon concentration in polar films is at the level of 10^20^ at/cm^3^, so around one order of magnitude higher than in the semi-polar films. Similarly, hydrogen and nitrogen contents are three times higher in polar films than in semi-polar films. Interestingly, the differences in carbon concentration also persist after annealing, when the crystallographic structure considerably improves and crystallites are much larger, which suggests that carbon impurity is not only related to grain boundaries. A comparison of carbon concentration vs. T_g_ dependence with the same dependence of strain and microstrain shows a correlation between them for polar (001)–oriented ZnO films. The highest carbon concentration in these films appears at T_g_ = 200 °C, where the highest strain and the smallest crystallite size are observed. After RTA, when crystallite size increases fivefold and the stress is released, the carbon concentration drops by one order of magnitude. Such a considerable decrease of carbon concentration after annealing is not observed for other growth temperatures, though, also in this case, crystallite size strongly increases. This suggests that increased strain/microstrain in the polar ZnO layer might enhance carbon incorporation, however, correlation of this kind should not occur for semi-polar films, which show a generally lower impurity concentrations.

Hall effect measurements can easily differentiate between the electrical properties of both the as-grown and annealed ZnO_(002)_/*c*-Al_2_O_3_ and ZnO_(101)_/*a*-Al_2_O_3_ films in terms of resistivity, carrier density, and mobility for all T_g_. Electrical resistivity of the as-grown films ranges between 2 × 10^−3^ and 2 Ωcm. After RTA, the resistivity increases by 2–3 orders of magnitude and lies in the range of 1–10 Ωcm. This is related to the drop of electron concentration by 2–3 orders of magnitude, which can be assigned to a significant reduction of hydrogen concentration and the improvement of structural quality expressed in a larger crystallite sizes.

At the growth temperature of 160 °C, the charge carriers in both the as-grown ZnO/*c*-Al_2_O_3_ and ZnO/*a*-Al_2_O_3_ films show a maximum mobility of 25–30 cm^2^/Vs. The as grown semi-polar films exhibit a higher electron mobility than the polar ones, despite lower crystallite size and higher dislocation density. Higher electron mobility in these layers might be related to lower impurity concentrations and/or higher strain and microstrain. It is worth noting that, after RTA, when microstrain in semi-polar films is mostly released, the mobility value drops, even though crystalline size increases and concentrations of hydrogen drops, while carbon concentration remains at a similar level. These factors indicate an occurrence of some link between mobility and microstrain, as both decrease after annealing.

In contrast, the polar films show higher mobility after RTA, leading to increase of crystallite size. Furthermore, in this case, we see a clear correlation between electron mobility and microstrain, which show very similar variations versus growth temperature. The latter observation supports a correspondence between electron mobility and microstrain.

## 5. Conclusions

Electrical transport measurements indicated that thin ALD-ZnO films showed a high sensitivity to growth conditions, as well as the growth direction of ZnO films. The comparison of polar or semi-polar ZnO films allowed us to conclude that carrier transport is highly dependent on ZnO orientation and annealing parameters. Carrier mobility is found to be influenced by many factors, such as the concentration of impurities, orientation of a film, and its structural characteristics, i.e., the crystallite size, strain, microstrain, etc.

The study presented here is expected to be of great relevance in future applications. It shows that when transparent conductive oxide applications are considered, the polar, highly (001)–oriented ZnO films should be applied, as they reveal the highest electron mobility and carrier concentration values. In ALD technology, such ZnO layers should be deposited at a temperature of 160 °C, because, at this growth temperature, the high electrical conductivity is accompanied by a low (at the nanometer scale) surface roughness. On the contrary, the semi-polar (101)–oriented films are preferred for obtaining p-type ZnO films, because concentrations of carbon and hydrogen impurities are considerably lower than in polar films.

## Figures and Tables

**Figure 1 materials-16-00151-f001:**
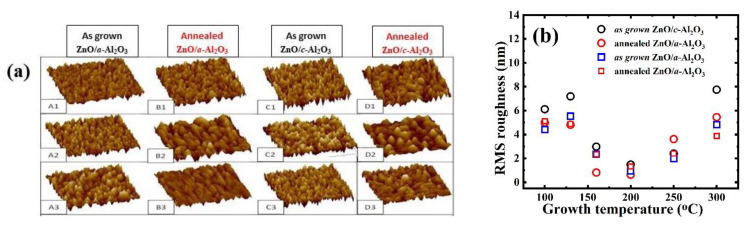
The 3D AFM images of as-grown and annealed (**a**) A and B column of ZnO/*a*-Al_2_O_3_ at T_g_ = 100, 130, 250 °C (A1–A3, as grown, B1–B3, annealed), (**b**) C and D column of ZnO/*c*-Al_2_O_3_ at T_g_ = 100, 130, 250 °C (C1–C3 as grown, D1–D3 annealed), with 10 × 10 µm scaling.

**Figure 2 materials-16-00151-f002:**
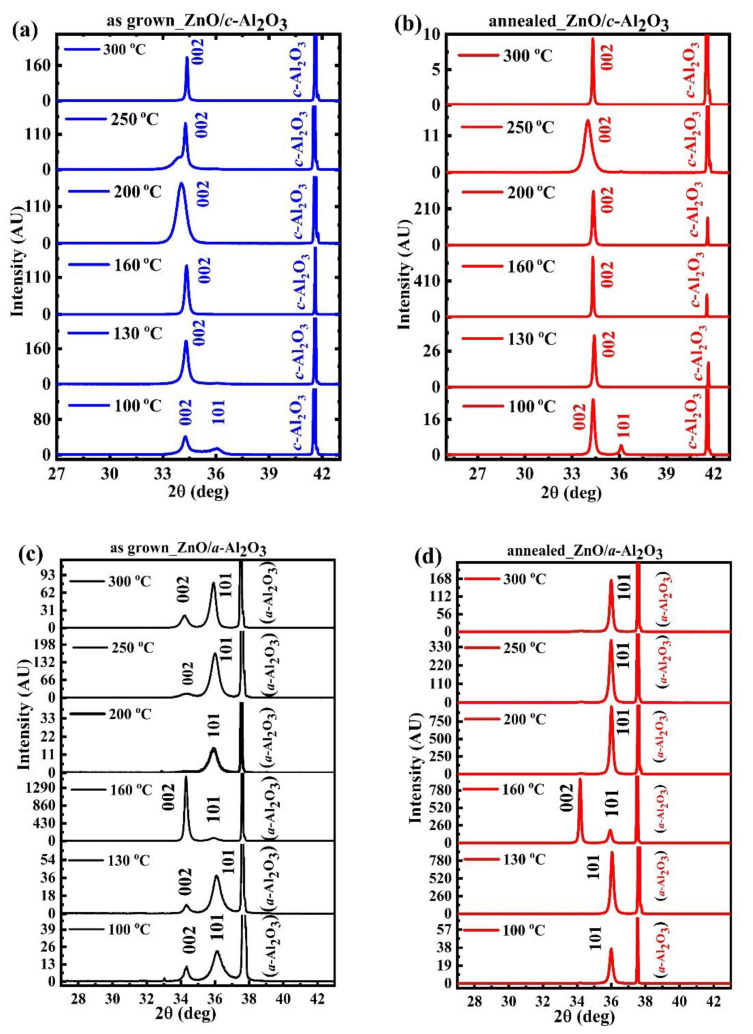
X-ray diffraction patterns of (**a**) as-grown and (**b**) annealed ZnO/*c*-Al_2_O_3_ films; (**c**) as-grown and (**d**) annealed ZnO/*a*-Al_2_O_3_ films with successive growth temperature ranges of 100 to 300 °C.

**Figure 3 materials-16-00151-f003:**
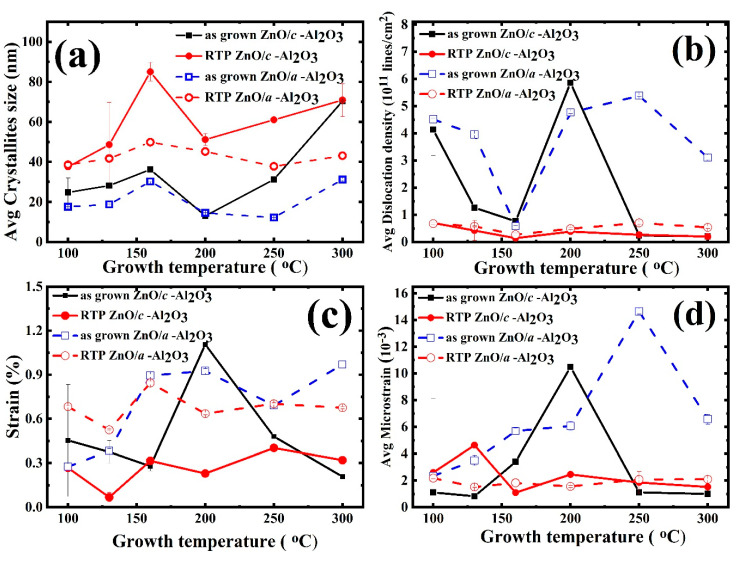
All microstructure parameter variations with T_g_: (**a**) crystallite size, (**b**) dislocation density, (**c**) strain%, (**d**) micro-strain, for as-grown and annealed ZnO films on *c*-Al_2_O_3_ and *a*-Al_2_O_3_. (The dotted line is for guideline).

**Figure 4 materials-16-00151-f004:**
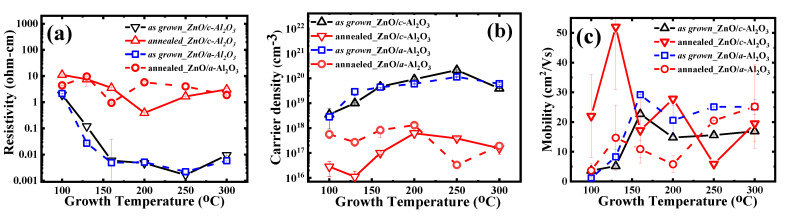
(**a**) Resistivity, (**b**) carrier concentrations, (**c**) mobility, with respect to T_g_ of as-grown and annealed ZnO films on *a*-Al_2_O_3_ and *c*-Al_2_O_3._ (Vertical lines show measurement uncertainty. In most of cases it is below the symbol size).

**Figure 5 materials-16-00151-f005:**
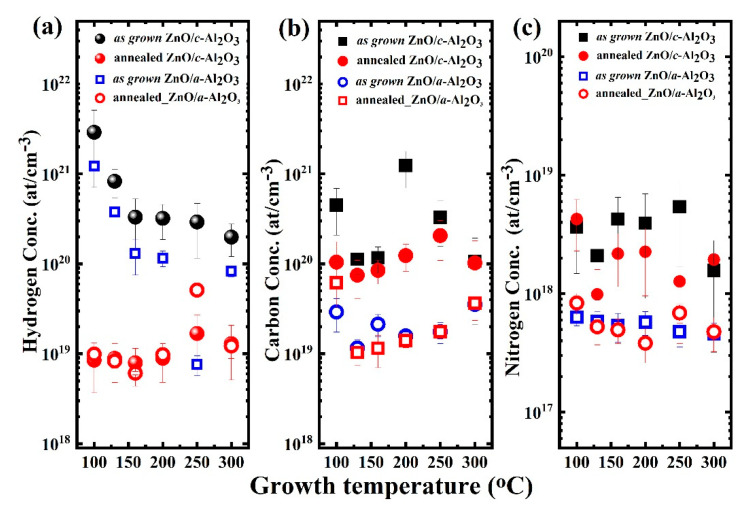
Impurities concentration incorporation: (**a**) hydrogen, (**b**) carbon, and (**c**) nitrogen, with successive growth temperatures T_g_ for as-grown and annealed on ZnO/*a*-Al_2_O_3_ and ZnO/*c*-Al_2_O_3_ films. (Vertical lines show measurement uncertainty).

**Table 1 materials-16-00151-t001:** Carrier density, mobility, and resistivity values of as-grown and annealed ZnO/*c*-Al_2_O_3_ films.

T_g_ (°C)	As-Grown Samples				Annealed Samples			
	ZnO/*c*-Al_2_O_3_				ZnO/*c*-Al_2_O_3_			
	(Thick ± RMS) (nm)	n_c_ (cm^−3^)	µ (cm^2^/Vs)	ρ (Ωcm)	(Thick ± RMS) (nm)	n_c_ (cm^−3^)	µ (cm^2^/Vs)	ρ (Ωcm)
100	150 ± 6	3.3 × 10^18^	3.65	1.92	151.1 ± 5	3.0 × 10^16^	22	11.1
130	162.5 ± 8	1.0 × 10^19^	5.2	1.2 × 10^−1^	165 ± 6	1.1 × 10^16^	52	7.53
160	158.5 ± 3	4.8 × 10^19^	22.7	5.87 × 10^−3^	161.7 ± 1	1.3 × 10^17^	17.2	3.53
200	135.7 ± 2	9.4 × 10^19^	14.8	4.69 × 10^−3^	137.4 ± 1	5.9 × 10^17^	27.8	3.89 × 10^−1^
250	114.4 ± 2	2.4 × 10^20^	15.6	1.65 × 10^−3^	116 ± 4	4.5 × 10^17^	5.88	1.66
300	123 ± 9	3.9 × 10^19^	16.9	9.4 × 10^−3^	128.4 ± 6	1.54 × 10^17^	19.5	3.07

**Table 2 materials-16-00151-t002:** Carrier density, mobility, and resistivity values of as-grown and annealed ZnO/*a*-Al_2_O_3_ films.

T_g_ (°C)	As-Grown Samples				Annealed Samples			
	ZnO/*a*-Al_2_O_3_				ZnO/*a*-Al_2_O_3_			
	(Thick ± RMS) (nm)	n_c_ (cm^−3^)	µ (cm^2^/Vs)	ρ (Ωcm)	(Thick ± RMS) (nm)	n_c_ (cm^−3^)	µ (cm^2^/Vs)	ρ (Ωcm)
100	135 ± 4	2.8 × 10^18^	1.2	2.15	139.5 ± 5	5.61 × 10^17^	3.7	4.39
130	160 ± 5	2.9 × 10^19^	8.3	2.7 × 10^−2^	161 ± 5	2.7 × 10^17^	14.7	9.62
160	140 ± 2	4.4 × 10^19^	29.2	4.86 × 10^−3^	151 ± 3	8.2 × 10^17^	10.9	0.95
200	123 ± 1	5.9 × 10^19^	20.6	5.1 × 10^−3^	123 ± 1	1.3 × 10^18^	5.81	5.75
250	110 ± 2	1.1 × 10^20^	25.1	2.2 × 10^−3^	111.5 ± 3	3.3 × 10^16^	20.5	4.02
300	97.2 ± 5	6.0 × 10^19^	25	5.78 × 10^−3^	100 ± 5	1.91× 10^17^	25.2	1.87

## Data Availability

The data presented in this study are available on request from the corresponding author.

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
