# Peer review of "Electrical and Structural Properties of Semi-Polar-ZnO/a-Al2O3 and Polar-ZnO/c-Al2O3 Films: A Comparative Study"

_materials, 2022, doi:10.3390/ma16010151_

Round 1
Reviewer 1 Report
In this paper, the authors prepared ZnO thin films by atomic layer deposition (ALD) on a- (100) and c- (001) oriented Al2O3 substrates. In addition, they investigated the influence of crystallographic orientation, structural defects and concentration of impurities on the electrical properties of the ZnO/Al2O3 films. The English in the manuscript must be carefully revised. Additionally, from the total number of 28 references presented in this article, there are only two from the last 8 years. Moreover, in the reviewer opinion this is not a new study, taking into account that in the literature there is already published a study of the authors (Sushma Mishra et. al., Materials 2021, 14, 4048) with ZnO films grown on a- (100) oriented Al2O3 substrates by ALD, obtained in the same experimental conditions: “The ALD processes were performed in 1000 cycles with 20 ms pulse time for both precursors, while purging time (N2) for deionized water and DEZn was 20 s and 8 s, respectively. It should be noted that the ZnO films were deposited on both substrates together during the same ALD process. The series of samples were obtained at temperatures, Tg, of 100, 130, 160, 200, 250, and 300 â—¦C.”. Consequently, in the reviewer opinion the manuscript cannot be accepted for publication in the current form.
Author Response
Reviewer 1
In this paper, the authors prepared ZnO thin films by atomic layer deposition (ALD) on a- (100) and c- (001) oriented Al2O3 substrates. In addition, they investigated the influence of crystallographic orientation, structural defects and concentration of impurities on the electrical properties of the ZnO/Al2O3 films.
- The English in the manuscript must be carefully revised.
The paper has been thoroughly checked for language corrections.
- Additionally, from the total number of 28 references presented in this article, there are only two from the last 8 years.
We thank for this remark. The reference list has been supplemented and new references have been added to fill this gap. All added references are marked in red in the revised version of the manuscript.
- Moreover, in the reviewer opinion this is not a new study, taking into account that in the literature there is already published a study of the authors (Sushma Mishra et. al., Materials 2021, 14, 4048) with ZnO films grown on a- (100) oriented Al2O3substrates by ALD, obtained in the same experimental conditions: “The ALD processes were performed in 1000 cycles with 20 ms pulse time for both precursors, while purging time (N2) for deionized water and DEZn was 20 s and 8 s, respectively. It should be noted that the ZnO films were deposited on both substrates together during the same ALD process. The series of samples were obtained at temperatures, Tg, of 100, 130, 160, 200, 250, and 300 â—¦”. Consequently, in the reviewer opinion the manuscript cannot be accepted for publication in the current form.
Indeed, the current study was inspired by previous results comparing the structural and electrical properties of thin ZnO-ALD films deposited on Si(100) and a-Al2O3 substrates published in the journal Materials last year. In the previous study, we found out that the type of substrate affects the dislocation density and strain of polycrystalline thin ZnO films, and thus electrical the transport in these layers. However, as mentioned in the Materials paper, the electrical parameters obtained for Si(100) and a-Al2O3 substrates were not easy to compare, because, after annealing, mobility values in ZnO/Si films were at the level of 200-1000 cm2/Vs, which is much higher than before annealing as well as above the value reported for a single ZnO crystal (~300 cm2/Vs). Therefore, it was suspected that 3 min. annealing performed at 800oC, affects ZnO/Si interface, leading to the formation of 2D electron gas. This issue was discussed and, consequently, the mobility values were not shown in the Materials paper (see table 4 therein). In order to avoid this inconvenience, the present investigations were performed using the sapphire substrate with two orientations in a series of deposition processed performed under O-rich (100oC) and Zn-rich (300oC) conditions. Because of using insulated sapphire substrates, all measured electrical parameters are reliable and the impact of structural defects on electron concentrations and mobility values can be reliably compared before and after annealing.
It should be emphasized that despite the large number of papers describing the electrical properties of ZnO films, some issues are hotly debated. The current lively discussion concerns, among others, the influence of native defect-impurity complexes, which, according to current knowledge, determine the conductivity level of this material. The presented research fits into the above debate as it investigates electrical conductivity of a series of films grown under O-rich (100oC) and Zn-rich (300oC) conditions, i.e. with different formation energy of native defects. The present investigation shows that not only the growth conditions (O/Zn rich) determine the ZnO conductivity. The level of structural defects is also important, which indirectly indicates their role in the formation of hydrogen impurity-native point defect complexes.
We supplemented the Introduction part with appropriate explanation in order to make the motivation of the paper more clear.
Reviewer 2 Report
The manuscript entitled “The comparison of electrical and structural properties of semipolar ZnO/a-Al2O3 and polar-ZnO/c-Al2O3 films”. Some issues to be addressed will improve the quality of the manuscript. Therefore, I recommend this work could be published after the major revision
1. 1. Should the author mention the article's originality in the summary section of the manuscript?
2. The english composition requires many improvements. The authors should proofread the manuscript carefully to minimize grammatical errors.
3. All the references mentioned in the paper should be cited in the text or vice-versa.
4. The research that is based on copper oxides for solar water splitting has been examined extensively, and a great number of studies have been carried out. For the sake of the reader's comprehension, I would ask that the author include a table of comparisons..
5. Based on the information presented in this article, Authod should elaborate on the XRD explanation. Journal of King Saud University – Science Volume 32, Issue 4, June 2020, Pages 2397-2405
6. To enhance the strength of the manuscript and a broader readership range, some important references, needs to be incorporated as given below.Electrochimica Acta Volume 111, 30 November 2013, Pages 593-600; Appl. Phys. Lett. 100, 011901 (2012)

Author Response
Reviewer 2
The manuscript entitled “The comparison of electrical and structural properties of semipolar ZnO/a-Al2O3 and polar-ZnO/c-Al2O3 films”. Some issues to be addressed will improve the quality of the manuscript. Therefore, I recommend this work could be published after the major revision
- Should the author mention the article's originality in the summary section of the manuscript?
We thank the Reviewer for this remark. The summary section of the paper has been rewritten in order to emphasize the originality of the paper.
- The English composition requires many improvements. The authors should proofread the manuscript carefully to minimize grammatical errors.
The paper was checked for grammatical errors.
- All the references mentioned in the paper should be cited in the text or vice-versa.
Thanks for this remark. All the references have been double-checked.
- The research that is based on copper oxides for solar water splitting has been examined extensively, and a great number of studies have been carried out. For the sake of the reader's comprehension, I would ask that the author include a table of comparisons.
There is probably a mistake. We did not investigate any copper oxides.
- Based on the information presented in this article, Author should elaborate on the XRD explanation. Journal of King Saud University – Science Volume 32, Issue 4, June 2020, Pages 2397-2405
We have elaborated on the XRD explanation by adding information about the phases identification method and interpretation of the observed discrepancies of the measured values of 2Theta and lattice parameters of the samples compared with the literature data.
- To enhance the strength of the manuscript and a broader readership range, some important references, needs to be incorporated as given below. Electrochimica Acta Volume 111, 30 November 2013, Pages 593-600; Appl. Phys. Lett. 100, 011901 (2012)
We added the papers to the reference list.

Reviewer 3 Report
Thanks for providing me the opportunity to review the article 'The comparison of electrical and structural properties of semipolar- 1 ZnO/a-Al2O3 and polar-ZnO/c-Al2O3 films'
Following are my comments to the authors, which may help to improve the article.
Abstract
The results given in the abstract are very general, they should be specific and clear.
Literature review
The literature review is not state-of-the-art, demeaning the relevance of the article in the present context. I could find one citation from 2020 and none from 2021, 2022, and 2023, hence the introduction part has to be rewritten and the most recent references should be included.
After presenting the review of literature, the article needs to identify the gaps in material-processing-applications, which are missing. The novelty of the research is also missing.
Section headings
The section heading is not consistent with the journal template hence, these should be corrected.
Section II. (Growth details and experimental techniques)
The experimental design is not clear, and the choice of materials and processing parameters is not explained so as to why the parameters are selected. For e.g.
1. Why the temperature range of 100-300 oC was selected?
2. Why double exchange chemical reaction was used?
3. What are the objectives of the study?
4. What are the limitations of the study?
5. What outcomes in terms of research know-how and industrial applications are expected?
Results (Again the section headings are inconsistent with the template)
The article has presented and discussed the characterization part satisfactorily but why the characterization was done, is lacking. The results should be presented in and around the research design.
Regarding phase interpretation from XRD
How the phases were identified? Were they matched with high scores or similar data cards? What parameter was selected in matching the phases, was it interplanar spacing or 2Theta? What was the accuracy of matching? Was it up to 2 decimal places or less?
Regarding crystallite size, dislocation density, and strain:
Figure 2 should be relocated to the middle of the discussion or immediately after the introductory notes.
The interpretation and discussion on the crystallite size and dislocation density are satisfactory but here the authors need to cross-refer to some most recent articles to support their claim. Apply this to strain and microstrain as well and remove the basic explanation of strain.
Electrical Properties
Electrical properties are presented and discussed satisfactorily, the only thing missing is cross-referencing with recent references to support the claims and interpretation.
AFM and SIMS results are ok, suggested moving the AFM results to the beginning of the results.
Summary and Conclusions
This section, though relevant to the research, is extended too much, the summary part should be moved towards the end of the results, and discussion and broad conclusions emerging from the research should be given in the conclusions. Further, without the end application and usefulness and relevance of the observations, the study doesn’t contribute significantly to the scientific know-how.
Author Response
Reviewer 3
Thanks for providing me the opportunity to review the article 'The comparison of electrical and structural properties of semipolar- 1 ZnO/a-Al2O3 and polar-ZnO/c-Al2O3 films'
Following are my comments to the authors, which may help to improve the article.
Abstract The results given in the abstract are very general, they should be specific and clear.
Literature review The literature review is not state-of-the-art, demeaning the relevance of the article in the present context. I could find one citation from 2020 and none from 2021, 2022, and 2023, hence the introduction part has to be rewritten and the most recent references should be included.
After presenting the review of literature, the article needs to identify the gaps in material-processing-applications, which are missing. The novelty of the research is also missing.
This is a good point, thanks the Referee for pointing this out. The reference list has been supplemented and recently published papers have been added. Additionally, the Introduction part has been re-written and information about motivation and novelty of the study is placed in the revised version of the manuscript.
Section headings The section heading is not consistent with the journal template hence, these should be corrected.
The section headings are corrected to be consistent with the Materials journal style.
Section II. (Growth details and experimental techniques)
The experimental design is not clear, and the choice of materials and processing parameters is not explained so as to why the parameters are selected. For e.g.
- Why the temperature range of 100-300 oC was selected?
This growth temperature range was selected, because it is related to change growth conditions from O-rich to Zn-rich, which in turn influences formation energy of acceptor-like and donor-like native point defects. This explanation is added in the corrected manuscript, at the end of the Introduction.
- Why double exchange chemical reaction was used?
Double exchange chemical reaction between Diethylzinc and water is the most frequently used to produce ZnO films by atomic layer deposition.
- What are the objectives of the study?
The objective of the study is thoroughly explained in the Introduction part of the revised manuscript. All the changes are marked in red.
- What are the limitations of the study?
The limitation of the study are structural changes that appear when growth conditions are changed from O-rich to Zn-rich. Within the ALD technique, the change of growth conditions is realized by growth temperature, however, the growth temperature influences structural quality of the films. The obtained results should be taken with this reservation. Thank you for this question. We placed the appropriate comment in the summary part.
- What outcomes in terms of research know-how and industrial applications are expected?
Information about the relevance of the study for future applications was added in the abstract and in the Conclusion section.
Results (Again the section headings are inconsistent with the template)
The article has presented and discussed the characterization part satisfactorily but why the characterization was done, is lacking. The results should be presented in and around the research design.
Regarding phase interpretation from XRD
How the phases were identified? Were they matched with high scores or similar data cards?
The experimental X-ray diffraction peaks match well to the ZnO data from JCPDS, record 36-1451:(McMurdie, H. et al., Powder Diffraction, 1, 76 (1986), one of records of this database, representing high quality data as evaluated by the database). Namely, the matching is observed for the reflections corresponding to the (001) and (if relevant) (101) plane orientation. The observed minor discrepancies in diffraction line positions are due to strain, as discussed below.
What parameter was selected in matching the phases, was it interplanar spacing or 2Theta?
We selected 2Theta as matching parameter, when comparing to the reflections of the record 36-1451.
What was the accuracy of matching? Was it up to 2 decimal places or less? In interplanar spacing the matching is at 3rd decimal place, whereas in 2Theta units it is between 0.05 to 0.1 deg.
In interplanar spacing the matching is at 3rd decimal place, whereas in 2Theta units it is between 0.05 to 0.1 deg. Discrepancies of this kind are routinely observed in ZnO thin films – they are observed if the film is strained (see the discussion in ref. [Water, W. and Chu, S‑Y, Materials Letters 55 67– 72 (2002)]). In particular, Water and Chu quote the range of lattice parameter (5.2457 A to 5.3689 A) due to the strain (interplanar spacing being half of these values), resulting in 002 reflection position between 34.15 deg and 33.35 deg. In our case the strain range is smaller (lattice parameter range is from 5.20 Å to 5.29 Å, corresponding to the 2Theta range from 34.41 deg to 33.93 deg). Both, our data and the Water and Chu data are consistent with the reference data for (unstrained) powder ZnO: lattice parameter value 5.20661 Å and 2Theta position 34.422 deg [McMurdie et al. (1986)].
Regarding crystallite size, dislocation density, and strain: Figure 2 should be relocated to the middle of the discussion or immediately after the introductory notes.
According to the Referee comment, Figure 2 has been relocated to page 8.
The interpretation and discussion on the crystallite size and dislocation density are satisfactory but here the authors need to cross-refer to some most recent articles to support their claim. Apply this to strain and microstrain as well and remove the basic explanation of strain.
The results on structural properties have been referred to some recent papers and appropriate references are added in the corrected version of the manuscript. We have also removed the basic explanation of strain from the main text.
Electrical Properties
Electrical properties are presented and discussed satisfactorily, the only thing missing is cross-referencing with recent references to support the claims and interpretation.
AFM and SIMS results are ok, suggested moving the AFM results to the beginning of the results.
According to the Referee comment, the AFM data were moved to the beginning of the Results section.
Summary and Conclusions
This section, though relevant to the research, is extended too much, the summary part should be moved towards the end of the results, and discussion and broad conclusions emerging from the research should be given in the conclusions. Further, without the end application and usefulness and relevance of the observations, the study doesn’t contribute significantly to the scientific know-how.
We decided to divide the Summary paragraph. In the present version of the paper one can find the “Summary and discussion” section followed by “Conclusions”. Some remarks on possible usefulness and relevance of the paper was added at the end of the manuscript.

Round 2
Reviewer 2 Report
the author solve all comments very carefully, i recommend it to accept in present form
Reviewer 3 Report
The authors have addressed the comments satisfactorily.